# Understanding School-Aged Childhood Obesity of Body Mass Index: Application of the Social-Ecological Framework

**DOI:** 10.3390/children7090134

**Published:** 2020-09-13

**Authors:** Keeyoon Noh, Jihyun Jane Min

**Affiliations:** 1Department of History, Philosophy, and Social Sciences, Pittsburg State University, Pittsburg, KS 66762, USA; knoh@pittstate.edu; 2Thomas Jefferson Independent Day School, 3401 Newman Rd, Joplin, MO 64801, USA

**Keywords:** childhood obesity, social-ecological model, race/ethnicity, older mothers, family structure, physical activity, school environments

## Abstract

In order to understand the prevalence of school-aged childhood obesity in the United States and suggest better methods to prevent and treat the public health problem, we examined it with significant and identifiable factors within the social-ecological model. To investigate the association between social-ecological factors and child obesity/overweight (BMI), we used the 5th wave of the Fragile Families and Child Wellbeing Study. The dataset included information on 9-year-old children. The sample size for our study was 2054. We utilized multiple normal distributions for missing values and the Ordinary Least Square regression analysis. Black and Hispanic children were more likely to be obese/overweight than White children; children with higher physical activity were negatively associated with higher obesity; older mothers were more likely to be associated with children’s obesity; family structure was also significantly related to the likelihood of childhood obesity; finally, school environment was significantly associated with child obesity. To combat childhood obesity, more school physical activities should be implemented, such as increasing physical education opportunities as well as building more sizable playgrounds and accessible recreation facilities at school and in communities. School environments also should be pleasant and safe for children. Health practitioners need to assess home environments to intervene for children’s health.

## 1. Introduction

Obesity is a major public health concern in the prospering nation of the United States—one that is unparalleled in comparison to other countries [1]. In fact, childhood obesity has also increased drastically over time; between 1971 and 2000, the number increased more than 100% with an obesity rate in 2000 at 10.4% [2]. These numbers have continued to grow exponentially as one-third of American children are considered obese/overweight, and this has remained unchanged in the past decade [3]. Obese children not only suffer from serious health issues, but they are also more susceptible to other health conditions, such as type 2 diabetes and cardiovascular disease [2]. Obesity is also associated with future mental health problems [4]. Other physical problems can include a higher risk of hypertension, high cholesterol, and an increased probability of obesity in adulthood [3]. In addition, obese/overweight girls can grow up to be obese as women, and they are more likely to gain an unhealthy amount of weight during pregnancy compared to average-weight women [5]. Overall, obesity is a cyclical process that negatively affects children’s adulthood health. Unlike the general stereotype that obesity is a result of laziness or lack of self-care, significant factors have been identified that predict a child’s chances of developing obesity. Numerous scholars have found such factors that mostly affect child obesity. For instance, some studies [6,7] found that Black children were more likely to be obese than White children. Other studies [8,9] discovered that obesity was highly likely to be associated with socioeconomic status (SES), and other studies [10,11] found an association between child obesity and accumulative social-behavioral factors—housing insecurity, neighborhood poverty, food security, and maternal depression.

According to the social-ecological framework, even though children are affected by various systems—individual, family, community/school, and government levels—few studies [12] have aimed to understand child obesity with those surrounding systems as a whole. Many studies about child obesity have made efforts to find the most crucial single element in each level without considering the significant contributions of several elements in each level or in combinations of levels [12]. Such findings would be problematic if not regarding high correlations between each individual element [12]. Additionally, Ohri-Vachaspati et al. [12], in one of few studies that examined child obesity within the Social-Ecological model, had a relatively small/medium sample size (*n* = 560) and did not include school environment (such as school types), child’s physical activity at school, parent’s marital status, and health insurance (Medicare/aid) availability. It is crucial to understand child obesity within a complex interaction of a child’s surrounding systems. Therefore, to add to the current literature on the topic and further comprehend child obesity/overweight, our study particularly detects several elements in each level or in several levels that are significantly associated with child obesity/overweight. For this purpose, our research explores its purpose utilizing the 5th wave of the U.S. national dataset of the Fragile Family and Child Wellbeing Study (FFCW) (9-year-old children) by applying the Social-Ecological Framework.

To explore our study, this paper first summarizes previous studies of an independent element that affected child obesity/overweight within the Social-Ecological Framework in the literature review; secondly describes this study’s methods and results; and finally makes recommendations for elementary school educators, public health practitioners, and policymakers in the discussion.

## 2. Literature Review

### 2.1. Individual Level

At the individual level, a child’s gender, race/ethnicity, and physical activities significantly correlate with obesity/overweight.

#### 2.1.1. Race

Substantial studies have found a correlation between child race/ethnicity and obesity. For instance, the studies by Kimbro et al. [2] and Martinson et al. [7] used the FFCW dataset to discover that young Hispanic children were more likely to be obese than White children in the U.S. Most importantly, Kimbro et al. [2] concluded that the tendency to become obese probably resulted from cultural practices in which Hispanic mothers are less likely to be concerned if their children eat fatty foods or drink whole milk frequently. However, another study by Whitaker and Orzol [13] examining the U.S. urban preschool children using the FFCW data determined that, after adjusting for the family level (maternal education, household income, and children’s food security), there was no relationship between race/ethnicity and child obesity/overweight. Overall, previous studies have shown some conflicts when focusing on the predictor of race/ethnicity and child obesity. 

#### 2.1.2. Physical Activity

Obesity/overweight occurs when people unbalance energy, consuming too many calories but burning few calories/energies. Therefore, physical activity is necessary to increase energy expenditure, help stay in energy balance, and lose weight. Three functions of physical activity prevent obesity. First, it decreases total body fat and slows the development of abdominal obesity. Second, muscle-strengthening activities, in particular, increase the energy that burns body fat and makes it easier to control weight. Third, it reduces depression and anxiety and motivates individuals to exercise consistently [14].

Several studies found children who spend more time in sedentary activities (e.g., watching television or playing video games) became obese/overweight [15,16]. Furthermore, the tendency toward physical activities can be understood according to race/ethnicity and family’s SES backgrounds. First, Kimbro and his colleagues [2] found that White children were less likely to watch more than five hours of television a day compared to Black and Hispanic children. Furthermore, Black children went outdoors less frequently compared to White and Hispanic children. This study provides examples that reflect that minorities, such as Black and Hispanic children, may be more susceptible to becoming obese/overweight because they get less physical exercise than White children. Second, however, SES shows indirect associations with physical activities. Two studies from Furstenberg et al. [17] and Kimbro et al. [18] unearthed that children living in lower SES households were more likely to be involved in outdoor activities than their counterparts from higher SES households. Further, Kimbro et al. [18] explained that families living in public housing were likely to stay at home and have more time to supervise their children’s play. This situation would allow lower SES children to have unstructured time at home and play outdoors instead of enrolling in preschool or daycare.

#### 2.1.3. Gender

Considering the prevalence of U.S. child obesity, obesity levels among adolescents between 12 and 19 years and school-aged children between six and 11 years are 20.6% and 18.4%, respectively, which is higher than among preschool children between two and five years (13.9%). Most of all, school-aged boys (20.4%) had a higher obesity rate than girls (16.3%) [19], and adolescent boys (12–17 years) were about twice as likely to be obese/overweight as adolescent girls (*p* < 0.05) [20]. In the study by Wang et al. [20], the researchers found that adolescent boys were more likely to exceed energy intake, were self-perceived as underweight, underestimated their body weight, and were more satisfied with their physical activity level than girls; girls were more likely to practice weight loss management through diet and were self-perceived as overweight more often than boys. Additionally, there were stronger associations between the risk of childhood obesity/overweight in boys than dieting to lose weight in girls.

On the other hand, a study by Suglia et al. [11]—while controlling for sociodemographic variables—observed that five-year-old girls’ obesity had a more significant association with cumulative social risk factors (combining parents’ partner violence, food insecurity, housing insecurity, maternal depressive symptoms, maternal substance use, and father’s incarceration), and these girls showed a higher level of obesity than girls who did not have social risk factors. Furthermore, they found that girls who had more externalizing problems (e.g., aggressive, destructive behaviors) were more likely to be obese.

### 2.2. Family Level

At the family level, household income, parental health, parental education, and family structure are considered essential factors that influence childhood obesity.

#### 2.2.1. Household Income

Although the United States has experienced a reduced poverty rate since the 1990s, the poverty rate for children has steadily risen, and the percentage living in poverty has increased from 15.5% in 2000 to 17.8% in 2004 [21]. Additionally, the population in severe poverty tends to be overparented children, Blacks, and Hispanics [21]. Households with young children are four times more likely to have Tier-I poverty (deficit of ≥ USD 8000) incomes than other families. Black households are more than three times as likely to have Tier-I incomes, and Hispanic households are more than twice as likely to have Tier-I incomes [21]. The poor have a higher susceptibility to develop unhealthy lifestyles, including smoking, drinking alcoholic beverages, eating fast foods, and physical inactivity. As a result, practicing such unhealthy lifestyles causes health problems such as disease, mental illness, and obesity [21]. The most recent research investigation by Starkey and Revenson [9] noted that the impact of household income status on a child’s obesity supported the information above.

Interestingly, however, the study by Kimbro et al. [18] discovered that there was no linear effect; children from the poorest and wealthiest households had the lowest Body Mass Indexes (BMIs) as opposed to children from the middle-income households who had the highest BMIs. Furthermore, the study also found that children living in public housing in the poorest neighborhoods with higher levels of physical disorder were more likely to be involved in outdoor play, which could lead to weight control.

#### 2.2.2. Parental Health and Health Insurance

One of the characteristics that parental health includes is depression. Maternal depression is often studied to determine whether this health condition correlates with a child’s higher likelihood of obesity/overweight. However, the results vary significantly. A study by Suglia et al. [11] found that maternal depression, controlling for other factors, was shown to influence five-year-old girls’ likelihood to become obese. However, another study conducted by Pineros-Leano [4] gave another insight into the effects of maternal depression. Whereas Suglia’s study [11] measured maternal depression in their influence of obesity, Pineros-Leano [4] uncovered that, specifically, maternal depression had no association with childhood obesity for children at ages of one and three. In the study conducted by Pineros-Leano [4], maternal depression instead was seen to be a trigger of long-term chronic childhood obesity (the children were considered obese at age five and later at age nine).

A study by Ohri-Vachaspati et al. [12]—measuring if parents who were obese/overweight influenced their child’s BMI—found that a child was more likely to become obese if the parent was also obese/overweight. This result was supported by another study conducted by Kimbro et al. [2] in which having an obese mother produced twice the odds of the child becoming obese/overweight for three-year-old children; however, this correlation was only seen for obese mothers rather than overweight mothers. Additionally, not having health care insurance at home significantly and negatively influences one’s ability to access medical care and community health services, which in turn increases the high risk of a child’s physical and mental health problems, including obesity [1,22].

#### 2.2.3. Parents’ Education

A parent’s education status has often been proven to be a barrier against childhood obesity. For mothers who achieved higher education, their children were less likely to become obese [12]. Nazarov and Rendall [23] concluded that mothers with higher education were more likely to have higher-paying jobs and would have more knowledge of healthier, more nutritious diets and physical activities, which leads to the prevention of childhood obesity. Another study by Baughcum et al. [24] explained that mothers with lower education were more likely to fail to perceive child obesity and easily overlooked their child’s health.

#### 2.2.4. Family Structure

Parental marital status significantly is related to a child’s obesity. A study by Schmeer [25], aiming to assess family structure and transition, found that these conditions were significantly correlated with child obesity/overweight. For instance, children whose mothers divorced or who had been living without partners for at least two years had a higher risk of becoming obese/overweight than children living in households with stable married parents [25]. Family transitions correlated as triggers, which caused stress and chaos, reduced resources and support at home, and hindered healthy eating, exercise, and sleep routines, which in turn increased children’s BMI and weight [25]. Additionally, Bzostek and Beck [26] discovered that a higher risk for child obesity/overweight could be found in households where children had unstable cohabiting mothers compared to stable cohabiting or stable married mothers. However, the researchers were unable to locate statistical differences between stable single and unstable single mothers. This result stressed that not all family structure transitions would be associated with an increased risk of childhood obesity [26]. Overall, maternal union transition could be significantly associated with child BMI because of SES, minority status, or health status before the transition rather than changes in the family environment [25].

#### 2.2.5. Parents’ Age

As opposed to numerous studies investigating parents’ educational levels and health conditions for child obesity/overweight, some studies specifically considered a mothers’ age when examining child obesity. Usually, mothers’ age has been treated as a controlled variable. One of the few studies which focused on maternal age and child health discovered that early motherhood has been associated with adverse developmental outcomes for both mothers and children because these mothers have little skill and experience to nurture their children well [27]. Agnafors et al. [28] argued that the mother’s age itself did not influence mothers and their children. Instead, young mothers tended to be associated with lower SES levels that negatively affect caring for children [28]. Another study by Chambers et al. [29] employing the FFCW data observed that older mothers were more likely to have high levels of household instability than younger mothers and were more likely to be associated with obesity/overweight. These findings could be possible predictors of older mothers having a higher chance of risk factors that can trigger child obesity [29].

### 2.3. Community/Neighborhood Level

Good communities that include safe and high-quality schools, as well as recreational amenities, positively affect children’s health [30]. Poor neighborhoods can cause residents and children to have physical disorders as they are not able to engage in physical activities within the community (e.g., outdoor play) [18]. Spending time in such deprived communities would lead children, in particular, to spend more time inside of houses/schools, which in turn increases the risk of obesity [18]. Good weight and healthy behavior can be the product of physical activity, active commuting from/to school, sports participation, and accessibility of recreations in communities [29,31]. A recent study conducted by Hinojosa et al. [32] further discovered that school-neighborhood environments were the most important contributions to child obesity/overweight. Negative environments—including those which are socioeconomically disadvantaged with high levels of a violent and dangerous crime, and fewer opportunities for physical education—are negatively associated with obesity/overweight. Public schools were considered good places to address weight-related behaviors for youth and children, and to increase physical and behavioral health, school interventions focusing on physical activity and monitoring school environment to make it safe were determined as necessary.

In addition to the studies that reported a possible mediating effect of physical activity on the relationship between school environments and the risk of obesity, other studies also found an effect of environmental and psychological stressors on the risk of obesity [33,34]. They found that stressors from a poor school climate (e.g., bullying, physical safety, and school connectedness) were associated with the risk of obesity among females.

As aforementioned in the literature review, it could be comprehensible that an independent variable or element could affect child obesity/overweight on different levels. Upon building up this information, the present study would be unique in that we will expand antecedents across multiple levels by questioning what will be significant and positive/negative variable(s) in individual, family, and community levels that affect child obesity/overweight.

## 3. Methods

### 3.1. Data and Sampling Process

This study used the public-use dataset from the 5th wave of the Fragile Families and Child Wellbeing Study (focal children’s ninth birthday). The data were collected from 20 large cities in the United States. There are six waves in the full data: Wave 1 (baseline), Wave 2 (Year 1), Wave 3 (Year 3), Wave 4 (Year 5), Wave 5 (Year 9), and Wave 6 (Year 15). We choose Wave 5 to look at the obesity/overweight of school-aged children. The 5th wave data were collected from August 2007 through April 2010. This includes interviews with core biological parents, primary caregivers, focal children, and teachers. A Home Visit assessment was also conducted, in which interviewers completed in-home observations of the home environment [35].

The survey components were administered by the primary caregiver (PCG) survey with computer-assisted telephone interviewing (CATI) followed by the core biological parent interviews. Home visits were scheduled during the primary caregiver and core biological parents’ phone interviews. The primary caregiver completed self-administered questions, and height (focal child only) and weight (focal child and biological mother) measurements were taken. As an incentive, families were given compensation for their participation in the nine-year wave of data collection: USD 25 for a primary caregiver, USD 30 for a biological mother, USD 75 for biological father, USD 30 for a child survey, and USD 65 for home visit activities [35].

The number of observations in the current dataset we chose is 4898. Out of 4898, for our study sample selection, we excluded 2644 cases in “not in the wave.” Out of the 2254 remaining cases, we also deleted cases who responded with “skip,” “don’t know,” or “refuse,” which ended in the valid sample of 2054. Finally, we used listwise deletion to handle missing values (*n* = 354), which ended in 1700 for the final sample used in the analysis, about 75 percent of the remaining sample of 2254. To deal with 25% of deleted cases properly, we used a common technique for handling missing data using Stata 14.2. We chose a multiple imputation (MI) technique using a multivariate normal distribution (MVN) over other alternative methods (e.g., listwise deletion, pairwise deletion, mean imputation, and single or deterministic imputation) because other methods introduce more bias into the parameter estimates than the MI method, and, therefore, produce inefficient results. Previous studies that employed Markov Chain Monte Carlo (MCMC) procedures have shown that an assumption of an MVN distribution resulted in reliable estimates even when the normality assumption is violated [36].

We specified the number of imputations as 11, although there is no recommended number of imputations. Recently, researchers recommend using larger values of imputations when the proportion of missing data is relatively high [37]. Furthermore, there is an advantage of using a larger value of imputations over using a smaller value of imputations. White, Royston, and Wood [38] found that estimates of regression coefficients and standard errors were significantly reduced and resulted in a better reproducibility. In this sense, researchers suggest that the number of imputations is equal to the highest proportion of missing cases or the highest fraction of missing information (FMI). In our case, the highest percentage of missing cases was 10.5. Therefore, we used 11 as the number of imputations.

We used all the variables in the imputation model plus auxiliary variables indicating the respondents’ ratio of total household income to the official poverty thresholds established by the U.S. In order to help the assumption of missing completely at random (MCAR) or missing at random (MAR). These auxiliary variables appear to predict the missingness of those variables that have missing cases. In addition, for the final research analysis, we used the Ordinary Least Square (OLS) regression analysis.

### 3.2. Measures

The following section focuses on the dependent variable, the child’s BMI, and the predictors of the study’s dependent variable.

#### 3.2.1. Dependent variable

We used the children’s measured BMI as the dependent variable in the study. According to the 5th wave of the Fragile Families and Child Wellbeing Study, the children’s BMIs were measured from height and weight measurements using a stadiometer during the home visit [35].

Although the BMI has been commonly used for assessing obesity/overweight in adults, adolescents, and children, the validity of the BMI for certain age groups has not been widely examined [39]. The Centers for Disease Control and Prevention (CDC) [40] provides age- and sex-specific measures, so-called BMI-for-age for assessing children’s obesity/overweight. As children’s body composition varies by age and varies by sex, it is important to use the BMI-for-age or BMI percentile measures to reflect the relativeness in assessing children’s obesity/overweight [40].

Some studies examined the validity of the BMI based on the measurement and the subject [41,42,43]. Karchynskaya et al. [41] assessed the validity of self-reported BMI and that of measured BMI from adolescents and found more accuracy using the measured BMI than the self-reported BMI. Some factors, such as underestimation due to the stigma of being obese/overweight from respondents, may affect the validity of self-reported BMI.

Given the purpose of the current study to investigate the association between social-ecological factors and childhood obesity/overweight rather than to assess a child’s obesity/overweight, the use of children’s BMI in the current study attenuates the measurement issues of BMI.

#### 3.2.2. Predictors

Individual level: For this level, we selected the child’s gender, race/ethnicity (White, Black, Hispanic, Other race), and physical activities at school. The variable of physical activities is continuous, from “a lot less active than most,” “a little less active than most,” “about the same as most,” “a little more active than most,” and “a lot more active than most.”

Family level: This level consists of the mother’s age, household income, family structure, whether the mother has health insurance coverage, and a mother’s health condition (depression). Two variables—mother’s age and household income—are continuous. The family structure includes married, separated, divorced, cohabiting, romantic not living together, and not romantic. Whether the mother has health insurance coverage is dichotomous. This was determined by the question, “Are you currently covered by any type of insurance, including private insurance, Medicaid, or another public, federal, or state assistance program which pays for medical care, or do you belong to a Medicaid HMO?” The answer to this question was either Yes (1) or No (0). The last variable was retrieved from the mother’s criteria of having depression, responding to mothers’ responses to either Yes (1) or No (0).

Community/Neighborhood level: There are two variables selected for this level: child’s school type and school’s neighborhood type. Child’s school type includes: regular public school, school with a magnet program, school of choice, and special education school. School’s neighborhood type was determined by the question, “how much of a problem is a crime in the school neighborhood?” The answers were marked by a big problem (1), somewhat of a problem (2), and no problem (3). These questions were answered by a focus child’s schoolteacher, and both of them were treated as categorical.

## 4. Results

Table 1 demonstrated the overall characteristics of the sample of children and families. In our sample, more than half of the children were boys (51.36%). Most of the sampled children were Black (51.07%), approximately a quarter were Hispanic (25.41%), almost a fifth were White (19.67%), and the remaining were in the Other category (3.9%). The school type varied among the sampled children as well. Many of the children attended a regular public school (87.2%), less than a tenth were in their school of choice (8.4%), several attended a school with magnet programs (4.09%), and very few were in a special education school (0.34%). Within these schools, the mean of the children’s physical activity was 3.11 (SD = 0.81). According to the children’s schoolteachers, the school environment was somewhat of a problem for the children (40.9%), not a problem (33.9%), and a big problem (25.3%). Among the mothers, the mean age was 34.25 years of age (SD = 5.93). Approximately 29% of the children’s mothers were married, 36.18% were cohabiting, 25.01% were in a romantic relationship but not living together, 7.98% were not in a romantic relationship, 1.49% were separated, and 0.36% were divorced. The mean household income was USD 43,892.84 (SD = 48,213.39). The only dependent variable in our study was a child’s BMI. According to our study, the mean BMI was 19.53 (SD = 4.47).

Table 2 indicated the bivariate analysis between all the variables in the model. A child’s BMI was correlated with a child’s gender and race/ethnicity, physical activity, household income, the school environment, and the family structure. We did not find an issue with multicollinearity among independent variables in the model; the variables with the highest Variance Inflation Factor (VIF) values were Black and Hispanic (2.38 and 2.03, respectively). The VIF values for the rest of the variables were lower than 2.0.

Table 3 showed both the results of the OLS regression model before (Model 1), and after (Model 2) we multiply imputation for missing cases. The noticeable difference between the two models was the number of observations due to the multiple imputations (*n* = 1700 in Model 1; *n* = 2054 in Model 2). Child’s race/ethnicity, physical activity, school environments, and the family structure were statistically significant in Model 1. In contrast, a child’s race/ethnicity, physical activity, school environments, mother’s age, and the family structure were statistically significant in Model 2. We then broke Model 2 into different models (Models 2A, 2B, and 2C, as shown in Table 4) based on the social-ecological framework.

Table 4 showed the results of the OLS regression models predicting a child’s BMI based on the social-ecological framework. While Model 2A only included the variables at the individual level, Models 2B and 2C included the variables at the family level and those at the community/neighborhood level, respectively. It was noteworthy that all the statistically significant predictors in Models 2A and 2B were also statistically significant in Model 2C with the same directions of the effects.

Model 2A indicated that a child’s race/ethnicity and physical activity were significant predictors of the child’s BMI. Racial/ethnic minority children (Black and Hispanic) were at higher risk of obesity/overweight than White children (*b* = 1.56; *p* < 0.001, *b* = 1.84; *p* < 0.001, respectively). As children had more physical activities, their risk of obesity/overweight was reduced (*b* = −1.06, *p* < 0.001).

Model 2B showed that a child’s race/ethnicity, physical activity, mother’s age, and family structure were significant predictors. Like the results from Model 2A, racial/ethnic minority children (Black and Hispanic) were at higher risk of obesity/overweight than White children (*b* = 1.22; *p* < 0.001, *b* = 1.70; *p* < 0.001, respectively), and more physical activities reduced the risk of obesity/overweight (*b* = −1.08, *p* < 0.001).

According to the results from our final model, Model 2C, a child’s race/ethnicity and mother’s age were positively associated with a child’s obesity/overweight, while physical activity was negatively associated. The family structure and school environments were also significantly associated with the child’s obesity/overweight. Black and Hispanic children were at a higher risk of obesity/overweight than White children (*b* = 0.96; *p* < 0.01, *b* = 1.48; *p* < 0.001, respectively). As children had more physical activities, their risk of obesity/overweight became lower (*b* = −1.08; *p* < 0.001). As the age of children’s mothers increases, their children were at a higher risk of obesity/overweight (*b* = 0.05; *p* < 0.01). Children whose mothers were cohabiting (i.e., mothers who were in a romantic relationship but not living together), or were not in a romantic relationship, were at a higher risk of obesity/overweight compared to those whose mothers were married (*b* = 0.95; *p* < 0.001, *b* = 0.88; *p* < 0.01, and *b* = 0.85; *p* < 0.05, respectively). Children whose school teachers did not perceive the school environment as a problem were at a lower risk of obesity/overweight than those whose school teachers perceived it as a big problem (*b* = −0.59; *p* < 0.05).

## 5. Discussion

When putting social-ecological variables in one model, our research answered our research question that a child’s race/ethnicity, child’s physical activity and the environment at school, mothers’ age, and the family structure were most significantly associated with a child’s BMI. With respect to race/ethnicity, our finding supports those of many studies that have indicated having minority status could be a negative factor to increase the risk of child obesity/overweight. Even though we could not conclude the significant trigger which caused nine-year-old Hispanic and Black children’s obesity/overweight, considering the aforementioned social-ecological theory and literature review, the racial/ethnic factor could be mostly negative when combined with economic depreciation. Economic status plays a vital role in the health disparities of people of color in the United States [22,25]. Hispanics and Black people are more likely to live at less than 50% of the poverty line than other racial groups [14]. However, this is probably not the most explicable case in our study because economic status was statistically controlled for in our model. A more plausible explanation is that, as a previous study [2] found, there are cultural differences in the diet pattern between White parents and minority parents, especially Black and Hispanic. Although they were found to be statistically significant in our results, the actual differences between White children and children of color were very small, given the standard deviation of the BMI in the sample (see Table 1).

Just as our study found, children’s physical activity matters for children’s healthy weight, which also supports previous studies as to the positive linear relationship between sedentary activities and child obesity/overweight. When considering our study outcome of the likelihood of Black and Hispanic children to be obese/overweight, the tendency of engaging in little physical activity might be another risk factor for these children of color. It is quite apparent that an increase in physical activities would lead to weight loss. More importantly, this finding shows that the effect of physical activity remains significant when various other variables are considered simultaneously. To reduce the risk of obesity/overweight, regardless of individual and family characteristics, children should engage in more physical activities.

Another of our study’s findings—fragile family structure was negatively associated with child obesity/overweight—strongly supports findings in the previous studies that children are at a higher risk of obesity/overweight when living with lower SES, unmarried mothers or mothers experiencing divorce [44,45]. Although our study was not able to detect significant differences between married parents and separated/divorced parents, it found significant differences between married parents and other types of fragile family structure (i.e., cohabiting, romantic but not living together, and not romantic). It appeared that the number of observations in the cases of separated and divorced parents was too small (see Table 1) to detect statistical relationships.

For the other finding of the significant and positive relationship between mother’s age and child obesity/overweight, our result does not support the study by Goodman et al. [27], in which younger mothers’ immaturity did not help a child’s health. Even though young mothers would display immaturity, because of their energy and awareness of the risk of obesity/overweight, they could be more participatory in children’s physical activity or recommend their children to play outside, which in turn reduces the children’s chances of being obese/overweight. In addition, according to Chambers et al. [29], older mothers were more likely to have household instability, which impairs the promotion of a healthy lifestyle in the home environment. Overall, older mothers’ instability could increase their children’s obesity/overweight.

Lastly, our finding of the effect of school environments did not support findings in the previous studies, which pointed out a possible mediating effect of physical activity on the relationship between school environment and childhood obesity [30]. Our bivariate analysis did not show a correlation between school environment and children’s physical activity. However, school environment was still associated with childhood obesity in the regression model after controlling for physical activity and other variables. This finding supports the previous studies which reported children who were stressed about their school environments were at a high risk of obesity [3,33]. Although our study was not able to find a mediating effect of physical activity from the school environment, physical activities and physical/emotional stressors appeared to play significant roles in children’s risk of obesity/overweight.

### 5.1. Implications

As discovered in our study, a couple of variables are significantly correlated with school-aged children’s obesity/overweight. We noticed that, considering all levels, child race/ethnicity, child’s physical activity at school, school environments, mother’s age, and the family structure are strongly significant factors. To promote school-aged children, school educators need to encourage students of color, in particular, to have more chances to be physically active at school. Additionally, according to Suglia [46], educators must continue in their endeavor of weight loss intervention among children by educating children’s parents on ways to promote a healthier lifestyle for their children.

Within the community, there should be good-quality recreation and facilities where children can spend more time outside. In order to help older mothers not to be associated with their children’s obesity/overweight, we recommend that community/local counselors visit their houses to assess family stability, such as whether the mothers have a high level of insecurity (stress from partner relationships and food insecurity, for example). This intervention would mostly help these families keep healthy lifestyles for child and mothers’ overall health.

Finally, our findings emphasized that racial disparities are related to health disparities of childhood obesity/overweight. This suggests a need for tailored interventions that consider cultural, dietary, and lifestyle issues that result from different racial/ethnic backgrounds. This issue would suggest that policymakers advocate for health care of racial/ethnic minority households by addressing culturally appropriate interventions and policies [22].

### 5.2. Limitations

Although this study found significant effects of socio-ecological factors on a child’s obesity/overweight, it had some limitations. First, the study employed a secondary data analysis from the 5th wave of the Fragile Families and Child Wellbeing Study (FFCW). Although we attempted to include main predictors from previous studies to investigate factors of child obesity/overweight, there were still other potential variables excluded from the study, such as eating habits, a possible explanation for child obesity/overweight. It should also be noted that the dataset only focuses on biological parents in explaining different forms of family structures as a static socio-ecological factor. Given there are changes in family structures since a child is born as a dynamic factor, current or intermittent family structures are necessary to be included in the analysis [47]. Future research needs to expand the focus on other potential factors related to childhood obesity/overweight.

Second, the original dataset of the FFCW is limited to childhood experiences and health issues in major metropolitan areas in the United States. As these children’s daily routines and life activities differ from those in other countries or rural areas in other regions in the U.S., our findings are not necessarily applicable to children outside the sampled areas. Replicate studies need to be conducted to include more representative samples from both urban and rural areas throughout the world.

Third, we could not find “perfect” auxiliary variables that were correlated with all other variables in the analysis, when used as auxiliary variables to find a correlation with missing variables. Our measure of imputed cases was based on the auxiliary variables indicating the ratio of total household income to the official poverty thresholds established by the U.S. These variables were moderately correlated with other variables. Although they did not have complete information to be correlated with every variable to be useful, they were still effective in reducing bias [48]. For future studies, researchers may need to find a better auxiliary variable to predict missingness in the dataset. This may be determined based on their knowledge of the data and subject of interest.

Fourth, the study used the FFCW dataset from 2007–2010. Although the data are the most recent data available from the FFCW, it may be viewed as outdated to analyze and explain the prevalence and the current situations of childhood obesity/overweight in the U.S. However, the FFCW dataset is one of the largest datasets that interviewed parents in various forms of marital status and their children, referred to as fragile families. As we aimed to examine the effect of factors at the family level along with other factors at the individual and community levels on childhood obesity/overweight, the data in the study were the best data available. Furthermore, according to the Centers for Disease Control and Prevention (CDC)’s 2017 National Center for Health Statistics (NCHS), the prevalence of obesity/overweight among youths in the U.S. has not significantly changed between 2003–2004 and 2013–2014 [19]. The unchanged situations in the prevalence of obesity/overweight among youths may also provide a rationale for the use of these data.

## 6. Conclusions

Childhood obesity/overweight is certainly a public health issue that requires much attention and intervention by public policymakers. We do believe that diet and exercise are crucial factors in influencing childhood obesity/overweight, but our research endeavors to emphasize how there are social determinant factors that cause certain families to face more resistance from keeping their child healthy than what other families would experience. Our research findings support this statement as factors such as race and family circumstances have had a statistically significant impact on obesity/overweight for the Wave-5 children in the Fragile Families and Child Wellbeing Study. Even though our study is a secondary national dataset and has its limitations, our research findings build upon the knowledge of educators, policymakers, and practitioners who are concerned about the childhood obesity/overweight crisis occurring in the United States.

## Figures and Tables

**Table 1 children-07-00134-t001:** Descriptive Characteristics of Variables.

Variables	*n* (%)	Mean (SD)
Child Gender	2054 (100.00)	
Boy	1055 (51.36)	
Girl	999 (48.64)	
Child Race	2054 (100.00)	
White	404 (19.67)	
Black	1049 (51.07)	
Hispanic	522 (25.41)	
Other	79 (3.85)	
School Types	2054 (100.00)	
Regular public	1791 (87.2)	
School with a magnet program	84 (4.09)	
School of choice	172 (8.37)	
Special education	7 (0.34)	
School Environments	1837 (100.00)	
A big problem	464 (25.26)	
Somewhat of a problem	751 (40.88)	
Not a problem	622 (33.86)	
Health Insurance Covered	1980 (100.00)	
Yes	456 (23.03)	
No	1524 (76.97)	
Family Structure	1943 (100.00)	
Married	563 (28.98)	
Separated	29 (1.49)	
Divorced	7 (0.36)	
Cohabiting	703 (36.18)	
Romantic but not living together	486 (25.01)	
Not romantic	155 (7.98)	
Mother’s Depression	1974 (100.00)	
Yes	344 (17.43)	
No	1630 (82.57)	
Physical Activity	2041	3.11 (0.81)
Mother’s Age	1980	34.25 (5.93)
Household Income	1977	43,892.84 (48,213.39)
BMI	2031	19.53 (4.47)

**Table 2 children-07-00134-t002:** Bivariate Analysis between Variables.

	(a)	(b)	(c)	(d)	(e)	(f)	(g)	(h)	(i)	(j)	(k)	(l)	(m)	(n)	(o)	(p)	(q)	(r)	(s)	(t)
(a) BMI	1.00																			
(b) Boy	−0.07 *	1.00																		
(c) Black	0.05 *	0.01	1.00																	
(d) Hispanic	0.09 *	−0.00	−0.60 *	1.00																
(e) Other	−0.03	0.02	−0.20 *	−0.12 *	1.00															
(f) Child Activity	−0.19 *	0.23 *	0.10 *	−0.07 *	−0.04	1.00														
(g) Mother’s Age	0.02	0.01	−0.06 *	−0.02	0.02	−0.02	1.00													
(h) Household Income	−0.10 *	0.05 *	−0.22 *	−0.08 *	0.07 *	−0.02	0.27 *	1.00												
(i) Magnet Program	0.00	−0.01	0.05 *	−0.03	0.01	0.00	0.01	−0.02	1.00											
(j) School of Choice	0.02	−0.04 *	0.13 *	−0.07 *	−0.02	0.01	0.01	−0.03	−0.06 *	1.00										
(k) Special Education	0.01	0.04	0.02	0.00	−0.01	0.07 *	−0.04	−0.04	−0.01	−0.02	1.00									
(l) Somewhat a Problem	0.08 *	−0.02	0.09 *	0.06 *	−0.03	−0.02	−0.01	−0.11 *	0.03	0.01	−0.02	1.00								
(m) No problem	−0.14 *	0.02	−0.25 *	−0.07 *	0.06 *	0.01	0.07 *	0.29 *	−0.03	−0.06 *	−0.01	−0.60 *	1.00							
(n) Health Insurance	−0.00	0.01	0.09 *	−0.18 *	0.03	0.02	0.05 *	0.14 *	0.03	0.05 *	0.01	−0.03	0.05 *	1.00						
(o) Separated	−0.02	0.04	0.02	0.02	−0.00	0.02	−0.03	−0.03	−0.00	−0.01	−0.01	−0.02	0.00	−0.00	1.00					
(p) Divorced	−0.03	0.02	−0.03	−0.02	−0.01	−0.03	0.01	0.02	0.03	−0.02	−0.00	−0.03	0.04	0.03	−0.01	1.00				
(q) Cohabiting	0.06 *	0.00	0.04	0.07 *	−0.02	0.04	−0.16 *	−0.14 *	−0.01	0.01	0.03	0.07 *	−0.12 *	−0.09 *	−0.09 *	−0.05 *	1.00			
(r) Romantic, not living	0.05 *	−0.01	0.25 *	−0.10 *	−0.04	0.03	−0.08 *	−0.17 *	0.02	0.01	−0.01	0.06 *	−0.14 *	0.03	−0.07 *	−0.03	−0.43 *	1.00		
(s) Not romantic	0.02	0.01	0.07 *	−0.06 *	−0.01	−0.01	−0.08 *	−0.12 *	0.02	0.01	−0.02	−0.02	−0.04	0.01	−0.04	−0.02	−0.22 *	−0.17 *	1.00	
(t) Depression	0.01	0.03	0.02	−0.02	−0.01	0.01	0.78 *	0.15 *	0.01	0.01	−0.02	0.01	0.02	0.00	0.04	−0.02	−0.02	0.00	0.04	1.00

* *p* < 0.05.

**Table 3 children-07-00134-t003:** Ordinary Least Square Regression Predicting BMI with Complete Cases and Multiply Imputed Cases.

	Model 1 (Complete Cases)			Model 2 (MVN Imputation)		
Variables	Coef.	SE	*p*-Value	Coef.	SE	*p*-Value
Gender (Girl)						
Boy	0.04	0.22	0.867	−0.15	0.20	0.456
Race/Ethnicity (White)						
Black	0.86	0.32	0.008	0.96	0.29	0.001
Hispanic	1.31	0.35	0.000	1.48	0.31	0.000
Other race	−0.31	0.58	0.586	0.30	0.54	0.580
Physical Activity	−1.12	0.13	0.000	−1.08	0.12	0.000
Mother’s Age	0.05	0.02	0.011	0.05	0.02	0.006
Household Income	−0.01	0.00	0.118	−0.00	0.00	0.737
School Type (Public School)						
School with magnet program	−0.18	0.52	0.731	0.06	0.49	0.900
School of choice	0.19	0.38	0.617	0.23	0.35	0.520
Special education	1.03	1.96	0.597	1.25	1.65	0.449
School Environments (A big problem)						
Somewhat of a problem	0.01	0.27	0.974	0.09	0.26	0.730
Not a problem	−0.61	0.30	0.047	−0.59	0.29	0.038
Health Insurance Covered (No)						
Yes	0.26	0.26	0.314	0.33	0.24	0.159
Family Structure (Married)						
Separated	0.45	0.93	0.624	−0.07	0.89	0.934
Divorced	−1.36	1.79	0.447	−1.29	1.66	0.436
Cohabiting	0.94	0.30	0.002	0.95	0.27	0.000
Romantic but not living together	1.01	0.33	0.002	0.88	0.31	0.005
Not romantic	0.89	0.45	0.051	0.85	0.41	0.039
Mother’s Depression (No)						
Yes	−0.05	0.28	0.849	0.11	0.26	0.665
Constant	28.67	3.87	0.000	27.92	3.45	0.000
*n*	1700			2054		
**F**	8.11 ***			8.74 ***		

Reference categories are in parentheses. SE = standard errors, *** *p* < 0.001.

**Table 4 children-07-00134-t004:** Ordinary Least Square Regression Predicting BMI in Social-Ecological Framework.

	Model 2A			Model 2B			Model 2C		
Variables	Coef.	SE	*p*-Value	Coef.	SE	*p*-Value	Coef.	SE	*p*-Value
Individual level									
Gender (Girl)									
Boy	−0.19	0.20	0.337	−0.16	0.20	0.422	−0.15	0.20	0.456
Race/Ethnicity (White)									
Black	1.56	0.26	0.000	1.22	0.28	0.000	0.96	0.29	0.001
Hispanic	1.84	0.29	0.000	1.70	0.30	0.000	1.48	0.31	0.000
Other race	0.56	0.54	0.299	0.42	0.54	0.436	0.30	0.54	0.580
Physical Activity	−1.06	0.12	0.000	−1.08	0.12	0.000	−1.08	0.12	0.000
Family level									
Mother’s Age				0.05	0.02	0.004	0.05	0.02	0.006
Household Income				−0.00	0.00	0.536	−0.00	0.00	0.737
Health Insurance Covered (No)									
Yes				0.34	0.24	0.149	0.33	0.24	0.159
Family Structure (Married)									
Separated				−0.09	0.89	0.924	−0.07	0.89	0.934
Divorced				−1.38	1.65	0.404	−1.29	1.66	0.436
Cohabiting				1.04	0.27	0.000	0.95	0.27	0.000
Romantic but not living together				0.97	0.31	0.002	0.88	0.31	0.005
Not romantic				0.92	0.41	0.026	0.85	0.41	0.039
Mother’s Depression (No)									
Yes				0.10	0.26	0.691	0.11	0.26	0.665
Community/Neighborhood level									
School Type (Public School)									
School with magnet program							0.06	0.49	0.900
School of choice							0.23	0.35	0.520
Special education							1.25	1.65	0.449
School Environments (A big problem)									
Somewhat of a problem							0.09	0.26	0.730
Not a problem							−0.59	0.29	0.038
Constant	32.46	1.57	0.000	28.43	2.87	0.000	27.92	3.45	0.000
*n*	2054			2054			2054		
**F**	26.46 ***			11.16 ***			8.74 ***		

Reference categories are in parentheses. SE = standard errors, *** *p* < 0.001.

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
