# Peer review of "Understanding School-Aged Childhood Obesity of Body Mass Index: Application of the Social-Ecological Framework"

_children, 2020, doi:10.3390/children7090134_

Round 1

Reviewer 1 Report

Thank you for the opportunity to review this article. The work is interesting, but some aspects should be taken into account.

Comments and suggestions for Authors:

First of all authors should add the tables to the manuscript. It is impossible to assess the result without tables. Therefore, I cannot evaluate the results now.

  1. Introduction:

It is incomprehensible for me to present such an extensive introduction with literature review. Authors should include a literature review in the introduction. The type “article” requires to clearly provide rationale for the study (with research questions and hypotheses) and clearly identify what this study adds to the current literature on this topic. Therefore this part should be shortened.

Chapter numbering is also wrong. Both introduction and Literature review are the “1” chapter. Also Parents’ Education has a 2.2.3. number, while Parental Health and Health Insurance has 1.1.1.

  1. Materials and Methods:
  • The study analyzes data from 2007-2010. These are data from 10 years ago. Looking at the dynamics of the increase in the prevalence of overweight and obesity among children and adolescents, these data cannot be related to the current situation.
  • The authors wrote about BMI percentiles, but nowhere is it written what percentile grids they used. Please describe it in detail in the methodology because this is what the research is about.
  1. Results:

As mentioned before, the results cannot be evaluated without the attached tables.

Reviewer 2 Report

This study examines the associations between various factors and the obesity status (Body Mass Index) of school-aged children, mainly focusing on school environment factors. Results from OLS analysis revealed that race, physical activities, family structure, and school environments are significantly associated with the obesity of children. I believe the paper has the potential to make a small but meaningful contribution to the literature. Yet, I have several significant concerns that would need to be addressed before I can comfortably recommend publication. These are not small issues, and I detail these concerns below in the hope that the authors will be able to find a way to address them.

One of the key arguments that the author(s) put forth is that school environments (ecological factor) serve as an important predictor of obesity of school-aged children. However, it is not clear how these theoretical factors (school environments) are related to obesity. Although the author(s) simply discusses previous literature and logic behind them, they failed to set the strong foundation of the relationship between school environments and the obesity of children. The author(s) should consider detailing and elaborating the theoretical associations between core predictors and the dependent variable in their literature review section.

In the method section, I have some methodological concerns. First, the author(s) did not mention anything about the data collection process. It would be helpful for the readership to know how information was captured. Especially, the author(s) stated that the data is nationally representative, and then it would be more reliable to provide information about the sampling process. More significantly, I do not think the “Fragile Families and Child Wellbeing Study” used a nationally representative sample since, as the study revealed on their website, this study targeted unmarried parents and their children. Please check the data and its sampling process.

Second, if there is no issue regarding the multicollinearity, Table 3 is unnecessary, and the author(s) can provide information about VIF values in the manuscript in a descriptive way.

Third, the dependent variable should be illustrated with its validity and reliability (i.e., previous literature) even though BMI is widely used to capture levels of obesity of children.

Fourth, I wonder whether the author(s) could break the “model 2” into different models. For now, the model 2 has all variables from different frameworks, which seems like smushed analyses. I recommend a hierarchical model. You may break down your models, for example (only suggestion), 

- Model 1: individual variables
- Model 2: individual predictors and family structure variables
- Model 3: individual, family, and environmental variables

Lastly, the quality of the writing needs substantially improved, which prevents the author(s)' paper from publication in its current form. The paper needs to be proofread and polished thoroughly. For example, on page 4, I believe it should be “was or is” not “us” -Parental marital status significantly us related to a child’s obesity- On page 9, some sentences are written in the different font sizes.

Round 2

Reviewer 1 Report

Thank you for the opportunity to review this resubmission.  Authors have done a nice job addressing reviewers' comments. Thank you. I am ok with acceptance.